# Colon-Targeted eNAMPT-Specific Peptide Systems for Treatment of DSS-Induced Acute and Chronic Colitis in Mouse

**DOI:** 10.3390/antiox11122376

**Published:** 2022-11-30

**Authors:** Jae-Sung Kim, Hyo Keun Kim, Minsoo Kim, Sein Jang, Euni Cho, Seok-Jun Mun, Joongho Lee, Dawon Hong, Seokhyun Yoon, Chul-Su Yang

**Affiliations:** 1Department of Bionano Technology, Hanyang University, Seoul 04673, Republic of Korea; 2Institute of Natural Science & Technology, Hanyang University, Ansan 15588, Republic of Korea; 3Department of Molecular and Life Science, Hanyang University, Ansan 15588, Republic of Korea; 4Center for Bionano Intelligence Education and Research, Ansan 15588, Republic of Korea; 5Department of Computer Science, College of SW Convergence, Dankook University, Yongin 16890, Republic of Korea; 6Laboratory of RNA Cell Biology, Graduate Department of Bioconvergence Engineering, Dankook University, Yongin 16890, Republic of Korea; 7Department of Electronics & Electrical Engineering, College of Engineering, Dankook University, Yongin 16890, Republic of Korea

**Keywords:** NAMPT, TLR4, CYBB, NLRP3 inflammasome, colitis

## Abstract

Nicotinamide phosphoribosyl transferase (NAMPT) is required to maintain the NAD^+^ pool, among which extracellular (e) NAMPT is associated with inflammation, mainly mediated by macrophages. However, the role of (e) NAMPT in inflammatory macrophages in ulcerative colitis is insufficiently understood. Here our analyses of single-cell RNA-seq data revealed that the levels of NAMPT and CYBB/NOX2 in macrophages were elevated in patients with colitis and in mouse models of acute and chronic colitis. These findings indicate the clinical significance of NAMPT and CYBB in colitis. Further, we found that eNAMPT directly binds the extracellular domains of CYBB and TLR4 in activated NLRP3 inflammasomes. Moreover, we developed a recombinant 12-residue TK peptide designated colon-targeted (CT)-conjugated multifunctional NAMPT (rCT-NAMPT), comprising CT as the colon-targeting moiety, which harbors the minimal essential residues required for CYBB/TLR4 binding. rCT-NAMPT effectively suppressed the severity of disease in DSS-induced acute and chronic colitis models through targeting the colon and inhibiting the interaction of NAMPT with CYBB or TLR4. Together, our data show that rCT-NAMPT may serve as an effective novel candidate therapeutic for colitis by modulating the NLRP3 inflammasome-mediated immune signaling system.

## 1. Introduction

Inflammatory bowel diseases (IBDs) including Crohn’s disease as well as ulcerative colitis (UC) are characterized by abnormalities in intestinal homeostasis, which result in chronic inflammation [1,2]. Biological drugs (anti-TNFα therapy and/or immune modulators) are effective for treating many people with IBD. However, many patients with severe disease fail to achieve remission because of a poor responsiveness to drugs, unavailability of mechanism-specific targeted therapy, or serious adverse effects requiring treatment [3,4]. Therefore, highly sensitive and specific predictive biomarkers along with novel therapeutic strategies are required to implement effective therapeutic strategies.

Nicotinamide phosphoribosyltransferase (NAMPT) acts as a catalyst of the rate-determining step of the nicotinamide adenine dinucleotide (NAD) recovery pathway and thus regulates intracellular NAD concentrations [5]. Many cell types secrete NAMPT, including monocytes and macrophages [6,7,8]. Accumulating evidence suggests that extracellular (e) NAMPT (visfatin) acts as a novel soluble factor with activities similar to those of cytokines, adipokines, and DAMPs [9,10,11]. Further, eNAMPT contributes to metabolic and inflammatory disorders such as obesity, diabetes, cancer, and particularly IBD [6,12,13,14]. The cellular and serum levels of eNAMPT are increased in patients with IBD who are unresponsive to anti-TNFα treatment (adalimumab or infliximab); and eNAMPT levels decrease in responsive patients to values comparable with those of healthy controls [15,16]. These findings support the conclusion that because eNAMPT levels correlate with a worse prognosis, this enzyme will serve as a potential target of drugs for effectively and safely treating IBD.

The effect of eNAMPT is mainly associated with the activation of inflammatory signals in macrophages, suggesting that the potent activation of TLR4-mediated NF-kB signaling occurs through direct binding to TLR4 [14,17,18,19]. In addition, changing intracellular NAD levels with NAMPT inhibitors or NAD precursors alters TLR4-mediated NF-kB activation and NLRP3 inflammasome activity, establishing a relationship between intracellular NAD levels and inflammation [20]. Further, eNAMPT may be associated with the activation of NADPH oxidase (NOX) to stimulate NOX-derived reactive oxygen species (ROS) through lipid-raft clustering, which in turn activates the NLRP3 inflammasome via TXNIP [21,22]. Therefore, activation of the NLRP3 inflammasome through the redox signaling pathway likely may occur through NF-kB signaling (priming) and interactions (activation) through the NAMPT-NOX association.

Intensive research focused on colonic-targeted drug delivery systems for topical treatment of colonic disease shows that they cause fewer systemic side effects and improve the effectiveness of oral delivery of therapeutic agents susceptible to acidic and enzymatic breakdown in the upper gastrointestinal tract [23,24]. A targeting moiety often enhances gene therapy. The inflamed colon can be targeted using a 12-residue peptide (TWYKIAFQRNRK; designated “CT” for **c**olonic-**t**argeting); it interacts with high affinity with the integrin α6β1, which is generated from the COOH-terminal globular domain belonging to the laminin-1 α1 chain [23,24]. Thus, the CT peptide may serve as a promising moiety for ameliorating colonic inflammation.

Our present data, which were acquired from genome-wide association studies of NAMPT in humans and mouse models of colitis, identified key driver genes in macrophages associated with IBD. Notably, we found that eNAMPT interacted with TLR4 or CYBB/NOX2 in IBD tissues. Moreover, recombinant multifunctional NAMPT (rCT-NAMPT) linked to the CT peptide that harbors the essential minimal residues required for binding CYBB and TLR4, may serve as a platform for developing safe and effective treatments of IBD.

## 2. Materials and Methdos

### 2.1. Analysis of Public Single-Cell RNA-Seq Data for Ulcerative Colitis

**Dataset used**: We used two datasets, SPC259 [25], available from the human cell atlas, and GSE182270 [26]. Table 1 shows a brief summary of the datasets. Although there are several datasets other than these two, e.g., GSE162335 [27] and GSE125527 [28], for ulcerative colitis, we chose these two since other datasets do not contain sufficient numbers of macrophage and T cells, which are our main focus in this study. Since the number of macrophages is limited in GSE182270, we used it only as supplementary data for DEG and GSE analysis to verify the main results from SCP259.

**DEG and GSEA**: We performed differentially expressed gene (DEG) analysis as well as gene set enrichment analysis (GSEA) individually on the selected cells for the two datasets. Specifically, macrophage, regulatory T cells, other CD4+ T cells and cytotoxic T cells were analyzed. The latest version of Seurat package [29] was used for the DEG analysis and FGSEA package [26,29] was used for GSEA. Since the number of cells in GSE182270 is much smaller than that in SCP259, we performed the analysis as follows.

(1)**DEG**: DEGs were identified separately for the two datasets with a *p*-value cutoff of 0.05. Rather than using an adjusted *p*-value, we used a non-adjusted one to obtain as many DEGs as possible.(2)**Selection of common DEGs**: We then selected those DEGs that have the same sign of log-fold **change** in the two datasets and then took the intersection of the two sets of DEGs, which is used for the GSEA.(3)**GSEAs with** common **DEGs**: GSEA analyses were performed with the selected DEGs with the log-fold changes for SCP259, since it might provide a better result when using a greater number of cells.

### 2.2. Mice, and Tissue-Culture

Wild-type C57BL/6 mice were supplied by Samtako Bio Korea (Osan, Korea). Primary bone marrow–derived macrophages (BMDMs) were harvested from mice and cultured in DMEM in the presence of M-CSF (R&D Systems, 416-ML) for 3–5 d, as described previously [30,31]. HEK293T cells (ATCC-11268; American Type Culture Collection) were cultured in in DMEM (Gibco) containing 10% FBS (Gibco), nonessential amino acids, sodium pyruvate, streptomycin (100 μg/mL) and penicillin G (100 IU/mL).

### 2.3. Reagents and Antibodies

LPS (*Escherichia coli* O111:B4, tlrl-eblps), Adenosine 5′-triphosphate (ATP, tlrl-atpl), Nigericin and DSS (Dextran Sulfate Sodium) were purchased from Invivogen.

Antibodies specific for NAMPT (ab236874) were purchased from Abcam. NAMPT (E-3), TLR4 (25), CYBB (54.1), CYBA (E-11), Actin (I-19), ASC (N-15-R), IL-18 (H-173-Y), caspase-1 p10 (M-20), HA (12CA5), Flag (D-8), GST(B-14), Myc (9E10), His (H-3) and V5 (C-9) were procured from Santa Cruz Biotechnology. Specific Abs against IkBα (L35A5) and Phospho-IkBα (14D4) were supplied by Cell Signaling Technology (Danvers, MA, USA). IL-1β (AF-401-NA) and NLRP3 (AG-20B-0014) were from R&D Systems and Adipogen, respectively.

### 2.4. Plasmid Construction

GST-NAMPT, Myc-NAMPT and Flag-TLR4 plasmids were sourced from Addgene. The full-length (FL) CYBB and mutant plasmids were previously described [32]. Plasmids encoding different regions of NAMPT (1-491, 10-116, 188-455, 456-491) were amplified by PCR from full-length NAMPT cDNA and subcloned into a pEBG derivative that codes for an N-terminal GST epitope tag flanked by the BamHI and NotI sites. Plasmids encoding different regions of TLR4 (1-811, 25-696, 717-811) were produced via PCR amplification from FL TLR4 cDNA and subcloned into a pEF derivative encoding a C-terminal Flag-tag between the BamHI and NotI sites. The pEBG-GST mammalian fusion vector and the pEF-IRES-Puro expression vector were used to create all transient constructs in mammalian cells. All constructions were sequenced using an ABI PRISM 377 automated DNA sequencer to ensure that they were 100% identical to the original sequence.

### 2.5. Peptides

Tat-conjugated NAMPT or NAMPT peptides were commercially produced and purified in acetate salt form to circumvent undesirable responses in the cells by Peptron (Daejeon, Republic of Korea). The endotoxin level was determined by the Limulus amebocyte lysate test (Charles River Endosafe^®^ Endochrome-K™, R1708K, Wilmington, MA, USA) and was less than 3–5 pg/mL at the concentrations of the peptides employed in experiments.

### 2.6. Recombinant Protein

To obtain recombinant rCT-NAMPT proteins, colon-targeting sequences of NAMPT amino acids (57–65), NAMPT amino acids (52–65), and CT peptides (TWYKIAFQRNRK) were cloned with an N-terminal 6xHis tag into the pRSFDuet-1 Vector (Novagen) and induced, harvested, and purified from *Escherichia coli* expression [30,31], in accordance with the standard protocols recommended by Novagen. The rVehicle, rCT, or rCT-NAMPT proteins were dialyzed through a permeable cellulose membrane and tested for lipopolysaccharide contamination using a Limulus amebocyte lysate assay (Bio-Whittaker) and contained <20 pg/mL at the concentrations of rVehicle, rCT, or rCT-NAMPT proteins used in the experiments.

### 2.7. GST Pulldown, Immunoblot, and Immunoprecipitation Analysis

293T and BMDMs were treated as specified and subjected to GST pull-down, Western blotting, and co-immunoprecipitation analysis, as previously reported [31,32,33,34]. 

293T cells were collected and lysed in NP-40 buffer supplemented with a full protease inhibitor cocktail (Roche, Basal, CH), for GST pull-down. After centrifugation, the supernatants were pre-cleared for 2 h using protein A/G beads at 4 °C. Pre-cleared lysates were combined with a 50% slurry of glutathione-conjugated Sepharose beads (Amersham Biosciences, Amersham, UK), and the binding reaction was incubated at 4 °C for 4 h. The precipitates were thoroughly rinsed with lysis buffer. Proteins bound to glutathione beads were eluted by boiling for 5 min in Sodium Dodecyl Sulfate (SDS) loading buffer. 

Cells were collected and lysed in NP-40 buffer supplemented with a full protease inhibitor cocktail (Roche, Basal, CH) for immunoprecipitation. Whole-cell lysates were immunoprecipitated with the indicated antibodies after pre-clearing with protein A/G agarose beads for 1 h at 4 °C. In general, 1 mL of cell lysates was treated with 1–4 µg of commercial antibody for 8 to 12 h at 4 °C. After 6 h of incubation with protein A/G agrose beads, the immunoprecipitates were thoroughly washed with lysis buffer and eluted with SDS loading buffer by boiling for 5 min. 

Polypeptides were separated by SDS–polyacrylamide gel electrophoresis and then transferred to a PVDF membrane for immunoblotting (IB) (Bio-Rad, Hercules, CA, USA). Achieving immunodetection required specific antibodies. Chemiluminescence (ECL; Millipore, Burlington, MA, USA) was used to visualize an antibody bound to a target, and a Vilber chemiluminescence analyzer was used to detect it (Fusion SL 3; Vilber Lourmat, Collégien, France).

### 2.8. Flow Cytometry

Flow cytometry was used to quantify intracellular ROS levels in cells cultivated in serum-free media and loaded with the redox-sensitive dyes 2 µM dihydroethidium (DHE for O^2−^; Calbiochem) or 1 µM 2′,7′-dichlorodihydrofluorescein diacetate (H2DCFDA for H_2_O_2_; Calbiochem) [32]. mAbs were incubated at 4 °C for 20–30 min to determine cell surface protein expression, and cells were fixed using Cytofix/Cytoperm Solution (BD Biosciences) and, in certain cases, followed by mAb incubation to detect intracellular proteins. [35]. The mAb clones that were used were the following: ITGA6 (GoH3, BD Pharmingen™) and ITGB1 (HM β1-1, BD Pharmingen™). The cells were washed completely and quickly with pulse spinning before being analyzed in a FACSCalibur (BD Biosciences, San Jose, CA, USA). CellQuest software (BD Biosciences) was used to visualize the data, and FlowJo software was used to analyze it (Tree Star, Ash-land, OR, USA). 

### 2.9. Enzyme-Linked Immunosorbent Assay

TNF-α, IL-6, IL-1β and IL-18 levels were measured in cell culture supernatants and mouse serum using the BD OptEIA ELISA system (BD Pharmingen). All assays were carried out exactly as the manufacturer instructed.

### 2.10. Lentiviral shRNA Production

For silencing murine ITGA6 and ITGB1 in primary cells, pLKO.1-based lentiviral CaMKKb shRNA constructs (sc-38952-SH) and LKB1 shRNA constructs (sc-35817-SH) were obtained from Santa Cruz Biotechnology. GIPZ Lentiviral Mouse Itga6 shRNA constructs (RMM4431-200328849, RMM4431-200410349 and RMM4431-200411464) and GIPZ Lentiviral Mouse Itgb1 shRNA constructs (RMM4431-200340920, RMM4431-200350229, RMM4431-200386104 and RMM4431-200400706) were procured from Open Biosystems. Lentiviruses were generated via transient transfection utilizing packaging plasmids (pMDLg/pRRE, pRSV-Rev, and pMD2.VSV-G, sourced from Addgene) after Lipofectamine 2000-mediated transient transfection into HEK293T cells, as reported previously. At 72 h after transfection, the virus-containing medium was collected and concentrated by ultracentrifugation. Lentiviral vector titration was calculated using 293T cells, and the resulting lentiviruses were transduced into BMDMs, as previously described [32]. 

### 2.11. In Vivo Lentivirus Transduction

As previously stated, concentrated lentiviral particles were frozen at 4 °C and diluted in PBS and polybrene (8 g/mL final concentration; Sigma) to give a dose of 1 × 10 [10] pfu in a 100 L injection volume [32]. Mice were intravenously injected with a lentivirus expressing nonspecific shRNA (shNS) or shRNA specific for ITGA6 (sh ITGA6) or ITGB1 (shITGB1) 2 times and then orally administered DSS (Acute or chronic Colitis) and rCT-NAMPT; then, desired experiments were performed. 

### 2.12. Mouse Model of Colitis

As previously described, DSS-induced acute or chronic colitis mouse models were developed using 6-week-old C57BL/6 female mice (Samtako, Osan, Republic of Korea) [36]. To assess the trigger of acute colitis, mice were treated with 3% (*w*/*v*) dextran sodium sulfate (molecular weight: 36,000–50,000 kDa, MP Biomedicals, Santa Ana, CA, USA) dissolved in drinking water that was given ad libitum. An acute colitis model was transduced with Lenti-shNS or Lenti-shITG virus (1 × 10^11^ pfu/kg) on days -7 and -14 via i.v. before DSS treatment. While the mice were treated with 3% DSS for 6 days, rVehicle, rCT or rCT-NAMPT (50 μg/kg) was i.p. injected 8 times. The survival of mice was tracked for 12 days; mortality was recorded for *n* = 15 mice per group. The survival of the mice model transduced with Lenti-shNS had similar effects to the WT control. A chronic colitis model was treated with 3% DSS for 7 days at 3 cycles and water for 14 days in the interval of the cycle. The rVehicle, rCT or rCT-NAMPT (50 μg/kg) was i.p. injected at 2 cycles together with DSS treatment. The survival of mice was followed for 9 weeks; mortality was recorded for *n* = 15 mice per group. To calculate causality, account for perturbations, and reduce bias, a randomization method was used to randomly assign mice to either a treatment group or a control group (or multiple intervention groups). The DSS solutions were freshly prepared every two days. The non-DSS-fed mice in the control group had access to sterile distilled water. The humane endpoint for body weight loss (euthanasia required) is 20% (as compared to the original body weight of an animal). Without an approved exception request, body weight loss could not exceed 20%.

### 2.13. Clinical Score and Histology

Body weight, occult or gross blood loss per rectum, and stool consistency were measured every other day during the colitis induction to obtain the clinical score. The clinical score was determined by two trained investigators who were not aware of the treatment. Mouse distal colon tissues were fixed in 10% formalin and embedded in paraffin for immunohistochemistry. Four-millimeter paraffin slices were cut and stained with hematoxylin and eosin (H&E). As previously mentioned, a board-certified pathologist (Dr. Min-Kyung Kim, Kim Min-Kyung Pathology Clinic, Seoul, Korea) independently scored each organ segment without prior knowledge of the therapy groups [36].

### 2.14. Statistical Analysis

All data are reported as mean ± SD and were analyzed using the Student’s *t*-test with a Bonferroni adjustment or ANOVA for multiple comparisons. The statistical software program SPSS (Version 12.0) was used to conduct the analyses (SPSS, Chicago, IL, USA). At p 0.05, differences were judged to be significant. Data for survival were graphed and analyzed using the Kaplan–Meier product limit method, with the log-rank (Man-tele-Cox) test for comparisons in GraphPad Prism (version 5.0, La Jolla, CA, USA).

## 3. Result

### 3.1. Overexpressed eNAMPT and CYBB May Cause Chronic Inflammation in IBD

Analysis of the datasets SCP259 and GSE182270 revealed that overexpressed NAMPT and CYBB (the latter also known as NOX2 and gp91^Phox^) may cause chronic activation of the NLRP3 inflammasome in UC (Figure 1A and Appendix A). The gastrointestinal macrophage participates in chronic inflammation in IBD, and we therefore used common macrophage-specific DEGs to check the pathways, utilizing the pathway view tool [37]. The NAMPT–CYBB interaction is associated with downstream inflammatory signaling [22]. eNAMPT stimulates the NOX-mediated redox regulatory pathway, and NAMPT associates with TLR4 [11,17,18], possibly leading to upregulation of NLRP3 and pro-IL1β expression via the NF-κB signaling pathway (Figure 1A). Overexpression of eNAMPT triggers the M1-skewed transcriptional program in macrophages [22]. These and our findings on the overexpression of NAMPT and CYBB in inflamed tissues of patients with UC led us to analyze single-cell RNA-seq data in more detail, focusing on the pathways associated with activation of the NLRP3 inflammasome through the interaction of eNAMPT with NOXs.

Table 2 shows the DEGs associated with NLRP3 inflammasome activation via the redox signaling pathway (activation), with focus on the NAMPT–NOX association and NF-κB signaling (priming). Violin plots (Figure 1B,C) show the expression of these genes in macrophages (Figure 1A). An increase in ROS via the NAMPT–NOX interaction is evident, because ROS-related genes such as SOD2, NEAT1, and HIF1A are overexpressed in inflamed UC samples (*p* < 0.05). The percentages of macrophages expressing NAMPT were similar in inflamed and control tissues, and their expression levels were higher in inflamed vs. control samples. Although the adjusted *p*-value for the fold-change in NAMPT expression in GSE182270 was not statistically significant, this may be explained by the relatively small number of macrophages.

The gene set enrichment results (Appendix A) show that in macrophages, many of the immune signals are upregulated in UC compared with those in the healthy colon. These signals include interleukin (IL)s, the innate immune system, Toll-like receptor cascades, TNFα signaling via NF-κB, inflammatory responses, cytokine signaling in the immune system, and IL-4 and IL-13 signaling (Figure 1A). These signaling events are involved in NLRP3 activation and in priming signaling pathways (Figure 1A). Further, genes associated with IFNγ signaling, antigen processing and presentation, as well as MHC Class II antigen presentation, were downregulated. Taken together, these data indicate that eNAMPT contributes to UC and that macrophages in UC transmit higher levels of innate immune signals associated with NAMPT–CYBB-driven signaling pathways.

### 3.2. eNAMPT Interacts with CYBB and TLR4

We further analyzed NAMPT and CYBB expression in colon tissue sections from normal subjects and patients with UC. The mRNA and protein levels of NAMPT and CYBB were approximately three to four times higher in the colon tissues of patients with UC compared with those of controls (Figure 2A). Further, the expression of NAMPT and CYBB markedly increased in association with increased disease severity in normal mice or those with acute or chronic colitis (Figure 2B). 

To confirm a role for eNAMPT in activating the NLRP3 inflammasome in macrophages, we examined whether the interaction of eNAMPT with TLR4 or CYBB was involved. A two-signal model was proposed to explain activation of the inflammasome by NLRP3 [22]. Signal 1 is a priming signal created by microbial components or endogenous cytokines that mediates NF-κB activation and subsequent increase of NLRP3 and prointerleukin-1β levels. Various molecules, including extracellular ATP and pore-forming toxins, transmit inflammatory activation signals (signal 2). Several molecular or cellular events activate the NLRP3 inflammasome, such as ion flux, mitochondrial dysfunction, ROS generation, and lysosomal damage [22,38].

First, we found that NAMPT expressed by TLR4 (LPS) or an NLRP3 inflammasome inducer (LPS/ATP) increased intracellular (i) NAMPT expression and markedly increased extracellular (e) NAMPT expression in macrophages (Figure 2C). Further, eNAMPT interacted with TLR4 or CYBB in the presence of LPS; and eNAMPT treated with recombinant NAMPT protein interacted with TLR4 or CYBB in macrophages. In contrast, the NAMPT interaction with CYBA was negligible in macrophages (Figure 2D). Further, an interaction between NAMPT and NLRP3 or ASC was undetectable (Figure 2E). Furthermore, the in vitro interaction between NAMPT and TLR4 or CYBB, as evaluated by a fluorescence binding experiment with recombinant proteins and fluorescently labeled TLR4 or CYBB with NAMPT, revealed a sufficiently high affinity (TLR4, 219 nM; CYBB, 896nM) (Figure 1). Together, the data suggest that the interaction of eNAMPT with TLR4 or CYBB mediates the activation of the NLRP3 inflammasome in inflammatory colitis, which indicates the potential clinical importance of these events.

### 3.3. NAMPT Amino Acid Sequence Essential for Binding TLR4 and CYBB

To identify the amino acid (aa) residues in NAMPT that interact with TLR4 and CYBB, and we constructed vectors that express full-length and mutant NAMPT, TLR4, and CYBB. NAMPT comprises N-terminal, middle (NAPRTase), and C-terminal domains (Figure 3A). To identify the domain required for the interaction between NAMPT and TLR4, we employed constructs tagged with GST or Myc–NAMPT and Flag–TLR4. In 293T cells, the N-terminus of NAMPT bound TLR4 and the extracellular leucine-rich region (LRR domain) of TLR4 were essential for the interaction with NAMPT (Figure 3A).

The cell-penetrating TAT peptide (GRKKRRQRRRPQ) overcomes the lipophilic barrier of cellular membranes and thus delivers large molecules as well as small particles into the cell where they exert their activities [39]. To investigate in detailed the sequence of the N-terminus of NAMPT, we constructed TAT–NAMPT peptides (separated by 20 aa) included in the stretch aa 10-116 of NAMPT. We incubated the TAT–NAMPT peptides with Myc–NAMPT and Flag–TLR4 expressed in 293T cells and immunoprecipitated (IP) the complexes with a Flag antibody. Treatment with TAT–NAMPT (aa 40–69) diminished the binding of NAMPT to TLR4, indicating that this region binds TLR4 (Figure 3B left). To determine the minimum aa sequence necessary for NAMPT–TLR4 binding, we utilized the PredictProtein software (https://predictprotein.org; accessed on 9 July 2022), which predicts elements of protein function and structure using database searches, homology-based inference, machine learning, and artificial intelligence. [40]. PredictProtein predicted that aa 57–65 of NAMPT were required for binding TLR4, consistent with findings that NAMPT–TLR4 binding in 293T cells was diminished by TAT–NAMPT (aa 57–65) in a concentration-dependent manner (Figure 3B right).

Next, we investigated the region of NAMPT that bound CYBB in 293T cells. The N-terminus of NAMPT bound CYBB, and extracellular domain 2 of CYBB was required for its interaction with NAMPT (Figure 3C). Further, aa 40–69 of NAMPT, the TLR4 binding residues, bound CYBB (Figure 3D left). PredictProtein predicted that NAMPT aa 52–56 as those that that bind CYBB, consistent with the decrease in NAMPT–CYBB binding in 293T cells co-expressing TAT–NAMPT (aa 52–56) in a concentration-dependent manner (Figure 3D right).

We next tested whether TAT–NAMPT aa 57–65 or aa 52–56 inhibited binding of eNAMPT to TLR4 or CYBB in macrophages. Consistent with the blockade of iNAMPT–TLR4 by TAT–NAMPT (aa 57–65) and iNAMPT–CYBB by TAT–NAMPT (aa 52–56), TAT–NAMPT (aa 57–65) specifically inhibited eNAMPT–TLR4 binding, and TAT–NAMPT (aa 52–56) specifically inhibited eNAMPT–CYBB binding (Figure 3E). Moreover, TAT–NAMPT (aa 52–65) inhibited the eNAMPT–TLR4 and eNAMPT–CYBB binding interactions. These findings show that NAMPT aa 57–65 and aa 52–56 are necessary for interacting with TLR4 or CYBB, respectively, demonstrating that NAMPT interactions with TLR4 and CYBB are genetically separate.

### 3.4. An NAMPT Peptide Inhibits Activation of the NLRP3 Inflammasome

To determine the effect of NAMPT peptides related to signal 1 for the NLRP3 inflammasome in macrophages, we exposed LPS-treated BMDMs to different concentrations of NAMPT peptides (aa 57–65, aa 52–56, or aa 52–65). NAMPT peptides aa 57–65 and aa 52–56 partially inhibited the activation of NF-κB, generation of ROS, and production of cytokines; however, NAMPT peptides aa 52–65 markedly inhibited the generation of ROS and NF-κB-induced cytokine production (Figure 4A–C). Further, TAT–NAMPT peptides aa 52–65, but not TAT–NAMPT peptides aa 57–65 and aa 52–56, partially inhibited NLRP3 inflammasome activation signal 1 (Appendix A). Thus, the TAT–NAMPT peptide was less likely to inhibit the interaction of eNAMPT with TLR4 or CYBB.

We therefore explored if NAMPT peptides have a special role in the modulation of signal-2 activation of the NLRP3 inflammasome. We found that NAMPT peptide aa 52–65 efficiently inhibited the maturation of IL-1β and IL-18, as well as ATP-induced caspase-1 cleavage and nigericin or DSS stimulation (Figure 4D upper, Figure 4E left and Appendix A). We found that the NAMPT peptide aa 52–65 specifically inhibited the action of eNAMPT (Figure 4D lower, Figure 4E right). These findings indicate that NAMPT’s actions in the TLR4-mediated signaling pathway and in CYBB-containing NOXs are functionally and genetically separable. Together, these data provide evidence that the NAMPT peptide aa 52–65 is an essential negative regulator of signal 1 and signal 2 in response to NLRP3 inflammasome activation.

### 3.5. Recombinant Multifunctional CT-NAMPT Protein Designed to Target the Pathologically Inflamed Colon

The experiments presented above indicate that NAMPT peptides (aa 57–65 and aa 52–56) directly blocked the binding of eNAMPT to the extracellular domains of CYBB and TLR4 and subsequently attenuated the activation of the NLRP3 inflammasome in macrophages. We used the peptide **T**WYKIAFQRNR**K** (designated TK), derived from the COOH-terminal globular domain of laminin-1 α1 chain, as a vehicle for targeted drug delivery to the colon. Thus, TK interacts with integrin α6β1, with high affinity for colonic tissue [23,41]. Accordingly, we developed a recombinant 12-residue TK peptide (CT) conjugated to a multifunctional NAMPT (rCT-NAMPT) in which CT targets the colon and harbors the essential and minimal aa residues required for CYBB/TLR4 binding. The authenticity of the predicted product was confirmed using SDS–polyacrylamide gel electrophoresis and immunoblotting (Figure 5A left). There were no significant differences compared with the vehicle control associated with rCT-NAMPT-induced cytotoxicity in BMDMs (Figure 5A right).

We employed a mouse model of DSS-induced colitis to further investigate the physiological significance of rCT-NAMPT in inflammatory colitis (Figure 5B,C). We found that the expression of ITGA6 (integrin α6) and ITGB1 (integrin β1), which bind to TK peptides (CT), was significantly increased in the colon of mice with acute colitis (Figure 5B). Next, to evaluate the specificity of rCT-NAMPT, we generated ITGA6- or ITGB1-knockdown mice through sh-Lentiviral transduction. The mice were treated with DSS and administered rCT-NAMPT/Cy5.5 via intraperitoneal injection on day 6. The rCT or rCT-NAMPT specifically targeted colonic tissues in mice with acute colitis, but not in other organs, in a concentration-dependent manner (Figure 5C and Appendix A, and data not shown). These results show that rCT binds to colonic tissues, which raises the possibility of designing colon-targeted drug delivery systems for use as pharmaceutical applications for treating DSS-induced acute colitis.

### 3.6. rCT-NAMPT Relieves Acute and Chronic DSS-Induced Colitis in a Mouse Model

We next evaluated the medicinal effects of rCT-NAMPT on mouse models in DSS-induced acute and chronic colitis. For this purpose, we generated ITG (ITGA6 and ITGB1)-knockdown mice through sh-Lentiviral transduction on days -7 and -14 via i.v. before DSS treatment. While the mice were treated with 3% DSS for 6 days, rVehicle, rCT or rCT-NAMPT (50 μg/kg) was i.p. injected 8 times. rCT-NAMPT significantly increased the survival rates of Lenti-shNS-transduced mice with DSS-induced colitis, but not those of Lenti-shITG-transduced mice. Neither rVehicle or rCT detectably affected mortality, suggesting that NAMPT peptides contribute to the regulation of the inflammatory response (Figure 6A). Furthermore, body-weight loss in Lenti-shNS-transduced mice treated with rCT-NAMPT was reduced by roughly 20% when compared to rVehicle- or rCT-treated mice. Body weights of Lenti-shITG-transduced mice were higher than in Lenti-shNS-transduced mice, but there was no significant difference (Figure 6B).

The colitis scores of mice were markedly decreased in Lenti-shNS-transduced mice treated with rCT-NAMPT (Figure 6C). After 12 days, we measured the length of the colon, which is an indication of colitis. Colon length recovered in rCT-NAMPT-treated Lenti-shNS-transduced mice, but remained unchanged in Lenti-shITG-transduced mice (Figure 6D). Further, we tested whether rCT-NAMPT exerted pharmacological activity in vivo. For example, the in vivo detection of interactions between the NAMPT and TLR4-mediated signaling pathways and CYBB-containing NOXs may prove to be important for the evaluation of rCT-NAMPT to identify drugs that treat lethal inflammatory disease. For this purpose, we analyzed the binding of NAMPT, ROS levels, and cytokine production in the colon. The interaction of NAMPT with TLR4 or CYBB was detected only in the colon of DSS-treated Lenti-shNS-transduced mice, but not in Lenti-shITG-transduced mice (Figure 6E). Cellular ROS levels decreased in Lenti-shNS-transduced mice treated with rCT-NAMPT, but not in Lenti-shITG-transduced mice (Figure 6F).

Moreover, the activity of myeloperoxidase involved in the production of TNF-α, IL-1β, IL-6, and IL-18 as well as the activation of NLRP3 inflammasome was measured (Figure 6G). The histological scores of colitis (H&E staining) revealed that the rVehicle- and the rCT-treated colon was disrupted by DSS in Lenti-shNS-transduced mice, and the addition of rCT-NAMPT restored the colonic barrier. Further, ITGA6/ITGB1-knockdown had no effect on rCT-NAMPT because of loss of colon-targeting ability (Figure 6H). Notably, rCT-NAMPT has a partially therapeutic effect against acute DSS-induced colitis in TLR4^−/−^ or CYBB^−/−^ mice compared with WT mice (Appendix A).

We then investigated the therapeutic effect of rCT-NAMPT on chronic colitis, subjecting mice to DSS and rCT-NAMPT for 66 days (Figure 7A). rCT-NAMPT, like acute colitis, boosted mouse survival rates by roughly 90%. (Figure 7B). Body weights fluctuated in the rVehicle- and rCT-treated mice, although rCT-NAMPT-treated mice maintained their body weights (Figure 7C). Further, the lengths of the colons of rCT-NAMPT-treated mice were restored compared with those of DSS-treated mice, which were significantly damaged (Figure 7D). H&E colon examination demonstrated that rCT-NAMPT dramatically improved colon vitality in DSS-treated mice (Figure 7E). These findings suggest that rCT-NAMPT has a therapeutic effect against DSS-induced colitis through inhibiting the NAMPT–TLR4 or –CYBB interaction in vivo.

## 4. Discussion

Here we show that rCT-NAMPT served as an effective novel candidate therapeutic for colitis by modulating the NLRP3 inflammasome-mediated immune signaling system through negative regulation of signals 1 and 2 in response to NLRP3-inflammasome activation. This finding represents a major paradigm shift in the treatment of colitis and an urgently needed therapeutic intervention. The main results of this study are as follows: (1) Analyses of two single-cell RNA-seq datasets, patients with colitis, and mouse models of acute and chronic colitis show that NAMPT significantly contributes to the pathology of UC. (2) Specifically, in macrophages, eNAMPT directly interacted with CYBB causing increased production of ROS and activation of the NLRP3 inflammasome in conjunction with the activation of the TNFα and NF-κB signaling pathways through direct stimulation by the eNAMPT–TLR4 interaction. (3) Amino acid residues aa 57–65 and aa 52–56 in NAMPT were essential for the interaction of NAMPT with TLR4 and CYBB, respectively. (4) A colon-targeting peptide designated TK was designed, and when conjugated to a multifunctional NAMPT peptide, targeted the inflamed colon in vivo. Further, the rCT-NAMPT-induced peptide inhibited NLRP3 inflammasome activation in vitro and in vivo. (5) rCT-NAMPT showed potential for treating acute and chronic DSS-induced colitis in mice. Collectively, these observations serve as a proof-of-concept for designing host-oriented therapeutic strategies and further demonstrate an engineered eNAMPT-induced NLRP3 inflammasome network associated with colitis.

The present data contribute several key insights into the role and therapeutic perspectives of macrophage-derived eNAMPT within the inflammatory response, particularly in the context of IBD. First, we found that macrophages distributed throughout the body, in contrast to T cells, serve as gatekeepers of tissue homeostasis and mediate innate and adaptive immune responses [42,43]. The present analyses of single-cell RNA-seq datasets show that compared to activated macrophages, T cells may be deactivated as suggested by (1) the overpopulation of regulatory T cells and (2) negative enrichment scores for most of the inflammatory signaling pathways in T cells. It is unclear if deactivation of T cells led to overactivated macrophages, or the opposite. Further, we are unable to explain why P2RY6 was not detectably or infrequently expressed in macrophages derived from inflamed UC samples. Nevertheless, blockade of the eNAMPT–CYBB interaction, reduction of ROS production, or both, in macrophages may ameliorate UC by relieving the activation of NLRP3 inflammasome. These two issues require further in-depth studies to enhance our knowledge of the pathogenesis of UC and other autoimmune diseases.

Second, this is the first report, to our knowledge, showing that macrophage-specific eNAMPT functions as a protective factor to alleviate the severity of colitis through inhibiting the binding by macrophages of CYBB and TLR4 to circulating eNAMPT. Recent studies have shown that NAMPT produced by macrophages binds to the C-C motif chemokine receptor type 5 in vivo, providing a stem cell activation niche that promotes muscle repair and regeneration [44]. Further, inhibition of NAMPT by the pharmacological inhibitor FK866 or genetic ablation of *Nampt* (*Nampt mKO*) may abrogate proliferation-inducing cues required for the injury-induced repair response in DSS-induced colitis [8]. However, as mentioned in this study [8], whether eNAMPT secreted by macrophages promotes colon repair and regeneration has not been determined. These findings [8], together with the comprehensive evidence of the present study, support the conclusion that macrophage-specific eNAMPT alleviates the severity of colitis. The preclinical work of Colombo et al. underlines that eNAMPT has a role in the etiology of some inflammatory illnesses via the effect of eNAMPT on macrophage inflammation. Furthermore, RNA-seq investigation of peritoneal macrophages reveals that eNAMPT activates an M1-distorted transcriptional program and preferentially promotes IFN-driven transcriptional activation in macrophages and human monocyte-derived macrophages via STAT1/3 phosphorylation [43].

Third, we show here that the eNAMPT–CYBB and –TLR4 axes contribute to the pathogenesis of colitis. Increased production of NOXs generates ROS, which are required for phagocytosis [8,32]. In this regard, our present findings highlight the important role of a metabolic axis composed of eNAMPT, NOX and ROS in maintaining the phagocytic activity of macrophages. Further, TLR4 serves as an alternative receptor for eNAMPT, as indicated by SPR studies and then confirmed by additional evidence [17]. For example, in human monocytes, a TLR4-neutralizing antibody reduces eNAMPT-mediated NF-kB activation [18].

Molecular docking studies on a minimized model of the NAMPT–TLR4 complex reveal a role for NAMPT-positive patches and the presence of stabilizing electrostatic interactions between K48, K68, K71, K73 with the negatively charged carboxyl moieties of TLR4 [40]. We show here that the amino acid residues 57–65 of NAMPT interacted with the LRR of TLR4. However, recent results demonstrate that eNAMPT exerts IFNγ-induced macrophage polarization independently of the TLR4-independent pathway [43]. KEGG analysis, a previous study, and our present data reveal the consistent enrichment of TLR signaling [45,46]. We therefore believe that it is reasonable to conclude that eNAMPT activity is TLR receptor- and ligand-specific and that the receptor may belong to the TLR family. Therefore, targeting these interactions may represent a promising therapeutic strategy designed to inhibit eNAMPT-induced inflammatory responses.

Fourth, rCT-NAMPT shows promise as a potential therapeutic. For example, numerous studies show that peptide-based biological agents have significant potential as immunotherapeutic agents against inflammatory diseases [30,31,33,34,35]. Colonic-targeted drug delivery systems for macromolecules may provide therapeutic benefits such as improved patient compliance (due to its painlessness and self-administration) and lower costs. pH-dependent systems, enzyme-inducing systems, receptor-mediated systems, and magnetically driven systems are among the strategies used to achieve more efficient colonic drug delivery for local or systemic effects [24,47].

Ligands are an essential component of targeted drug delivery systems, and the selection of ligands with high affinity is of considerable interest [23,24,47,48,49]. Here, we used TK peptides as suitable targeting ligands with potential applications for colon-targeted therapy [23]. Further studies must focus on different approaches to designing colonic-targeted drug delivery systems as well as on pharmaceutical applications and formulation technologies.

Notably, these rCT-NAMPTs do not fulfill the requirements of direct NAMPT enzymatic activities, which represent feasible alternatives to the conventional chemotherapy of colitis. Furthermore, the unknown specificities and selectivity of rCT-NAMPTs make it impossible to link their effects to host immune systems, and a shortcoming of an animal experimental model is whether it truly replicates human pathophysiology Other limitations include safety data, potential feasibility, unknown off-target effects, and pharmacokinetics for in vivo proof-of-concept studies. 

Our results show that the NAMPT, TLR4, and CYBB genes associated with mouse colitis share functions with those of humans. It is believed that molecular biology and immunological results from mouse model systems can be translated into insights into human biology and health through interpretation based on mouse experimental data. Therefore, additional studies are warranted to establish whether rCT-NAMPTs can be translated to the clinic and whether their effects can be confirmed in individuals with colitis.

In conclusion, to effectively maximize the anti-inflammatory effectiveness of immunotherapies targeted to the activated NLRP3 inflammasome, it is important to combine the specificity of the eNAMPT effect on macrophages as well as its efficacy for colonic targeting. The prospects for the treatment of colitis with next-generation peptide immunotherapeutic depend on the results of future translational and clinical trials.

## Figures and Tables

**Figure 1 antioxidants-11-02376-f001:**
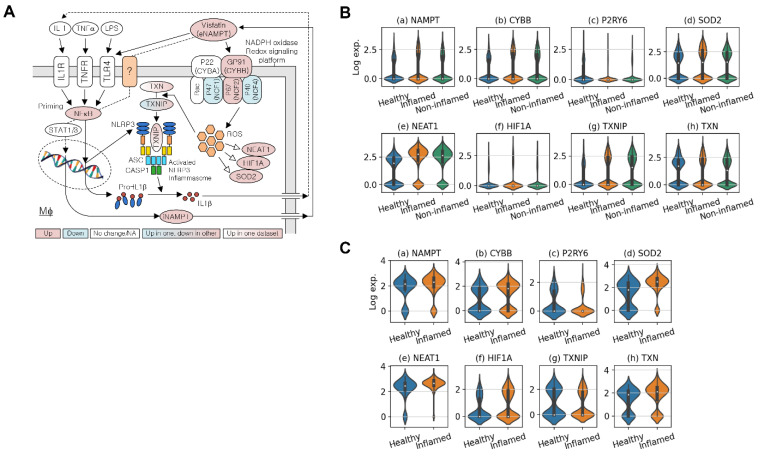
eNAMPT plays a key role in ulcerative colitis (**A**) NAMPT’s association with TLR4 or NADPH oxidase-driven NLRP3 inflammasome activation. (**B**) Violin plots for the expression of key genes in macrophages around NLRP3 inflammasome activation in SCP259. (**C**) Violin plots for the expression of key genes around NLRP3 inflammasome activation in GSE182270.

**Figure 2 antioxidants-11-02376-f002:**
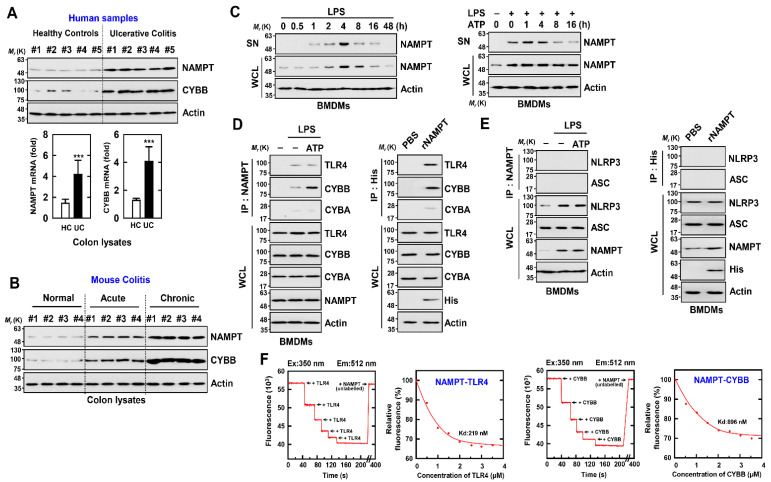
eNAMPT directly binds with CYBB or TLR4 (**A**) Normal human cases alongside UC patients were employed for IB with αNAMPT and αCYBB. WCLs (whole cell lysates) were utilized for IB with αActin (upper). Representative gel images derived from five independent healthy controls and patients are shown. *NAMPT* mRNA and *CYBB* mRNA expression measured by quantitative real-time PCR (lower). Five of ten normal human and UC patients’ data are shown. Statistical significance was evaluated by the Student’s *t*-test coupled with the Bonferroni adjustment (*** *p* < 0.001) versus human normal. (**B**) Colon of normal, acute and chronic colitis mouse were analyzed for IB with αNAMPT and αCYBB. WCLs were used for IB with αActin. Biological replicates (*n* = 10) for each condition were performed. (**C**) BMDMs were activated with LPS (100 ng/mL) for the durations indicated (left), then primed with LPS (100 ng/mL) for 4 h before being stimulated with ATP (1 mM) for the times indicated (right). IB in supernatant (SN) with αNAMPT and WCL with αNAMPT or αActin. (**D**,**E**) BMDMs were primed with LPS and stimulated with ATP (left) or incubated with rNAMPT (1 μg/mL) for 2 h (right). BMDMs were treated with IP with αNAMPT or αHis, trailed by IB with αTLR4, αCYBB or αCYBA (**D**) and IB with αASC or αNLRP3 (**E**). WCLs were used for IB with αTLR4, αCYBB, αCYBA, αASC, αNLRP3 or αActin. (**F**) Titration of fluorescently labeled TLR4 or CYBB with unlabeled NAMPT, using curve fit analysis to determine Kd (219 and 896 nM). The data come from five independent experiments that yielded comparable results.

**Figure 3 antioxidants-11-02376-f003:**
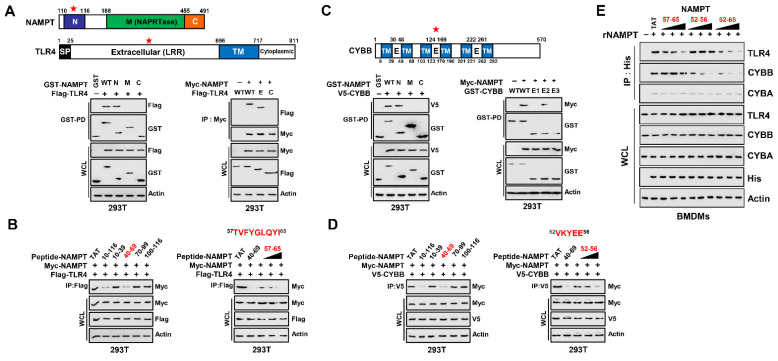
NAMPT (aa 57-65 and aa 52-65) is an essential region for binding to TLR4 and CYBB. (**A**) Schematic diagram of the structures of NAMPT and TLR4 (upper). The 293T cells were transfected with GST-NAMPT or its mutants and Flag-TLR4. 293T cells were subjected to GST pull-down, followed by IB with αFlag. WCLs were used for IB with αGST, αFlag or αActin (lower, left). The 293T cells were transfected with Flag-TLR4 or its mutants and Myc-NAMPT. The 293T cells were used for IP with αMyc, followed by IB with αFlag. WCLs were used for IB with αMyc, αFlag or αActin (lower, right). (**B**) 293T cells were transfected with Myc-NAMPT and Flag-TLR4 and treated NAMPT peptide or its mutants for 6 h (5 µM, left; 1, 5, 10 µM, right). The 293T cells were used for IP with αFlag, followed by IB with αMyc. WCLs were used for IB with αMyc, αFlag or αActin. (**C**) Schematic diagram of the structures of CYBB (upper). The 293T cells were transfected with GST-NAMPT or its mutants and V5-CYBB (lower, left). The 293T cells were transfected with GST-CYBB or its mutants and Myc-NAMPT (lower, right). The 293T cells were subjected to GST pull-down, followed by IB with αV5 or αMyc. WCLs were used for IB with αGST, αMyc, αV5 or αActin. (**D**) 293T cells were transfected with Myc-NAMPT and V5-CYBB and treated NAMPT peptide or its mutants for 6 h (5 µM, left; 1, 5, 10 µM, right). The 293T cells were used for IP with αV5, followed by IB with αMyc. WCLs were used for IB with αMyc, αFlag or αActin. (**E**) BMDMs were incubated with rNAMPT (1, 5, 10 μg/mL) for 2 hr. BMDMs were subjected to IP with αHis, followed by IB with αTLR4, αCYBB or αCYBA. WCLs were used for IB with αTLR4, αCYBB or αCYBA, αHis or αActin. The data come from seven independent experiments that yielded comparable results (**A**–**E**). Ref star: Binding domain with binding partners.

**Figure 4 antioxidants-11-02376-f004:**
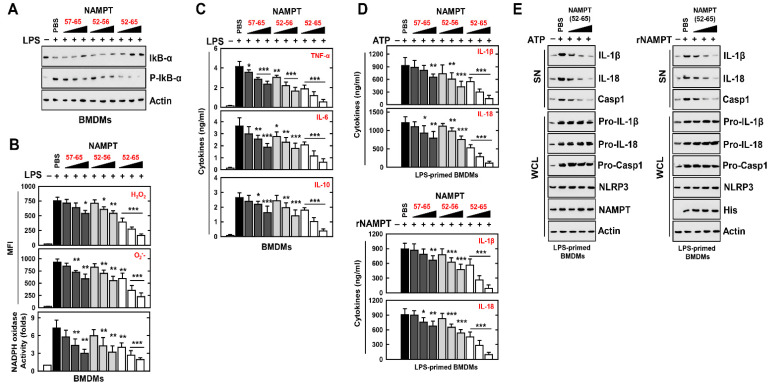
NAMPT peptides inhibit the signal 1 and 2 activation of the NLRP3 inflammasome. (**A**–**C**) BMDMs were stimulated with LPS for 4 h, pretreated with rNAMPT for 1 h (**A**,**B**) or 18 h (**C**). (**A**) WCLs were used for IB with αIkB-α, αP-IkB-α or αActin. (**B**) FACS analysis for H_2_O_2_ (probe for 2′, 7′-dichlorofluorescin diacetate) or O_2_- (probe for Dihydroethidium). Quantitative analysis of mean fluorescence intensities of H_2_O_2_ and O_2_- (upper). NADPH oxidase activity (lower). (**C**) Culture supernatants were harvested and analyzed for cytokine ELISA. (**D**,**E**) LPS-primed BMDMs were treated with rNAMPT for 1 h, and then activated with ATP for 30 min. (**D**) ELISA for IL-1β and IL-18. (**E**) IB in SN with αIL-1β p17, αIL-18 p18 or αCasp1 p10 and WCL with αPro-IL-1β, αPro-IL-18, αPro-Casp1, or αActin. The data come from seven independent experiments that yielded comparable results (**A**,**E**). The data come from three independent experiments that yielded comparable results. (**B**–**D**). Data are given as means ± SD of three experiments. Significant differences (* *p* < 0.05; ** *p* < 0.01; *** *p* < 0.001) versus LPS + PBS.

**Figure 5 antioxidants-11-02376-f005:**
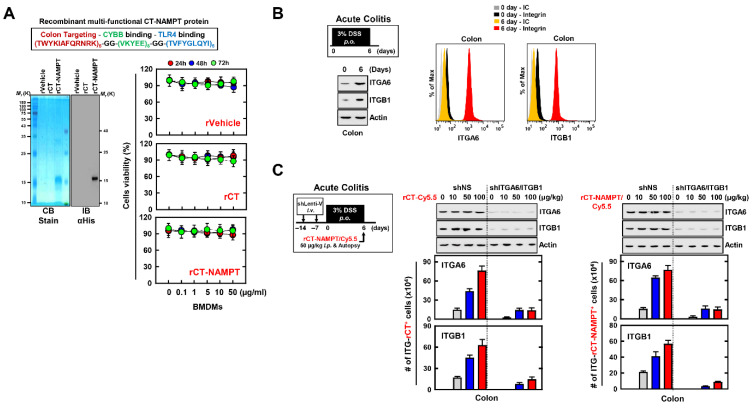
rCT-NAMPT targets the inflamed colon. (**A**) Schematic illustration of rCT-NAMPT (upper). Bacterially purified 6xHis-rCT-NAMPT, rVehicle or rCT were analyzed by Coomassie Blue staining or IB with αHis (lower, left). BMDMs were treated with rVehicle, rCT, or rCT-NAMPT for the periods and concentrations stated, and cell viability was determined using the MTT test (lower, right). (**B**) DSS-induced model for colitis. Mice were subjected to 3% DSS for 6 days and were evaluated at day 6 (left). Colon harvests were used for IB with αITGA6, αITGB1 or αActin and analysis of the number of ITGA6+ or ITGB1+ cells by FACS. (**C**) The scheme of the acute colitis model transduced with Lenti-shNS or Lenti-shITG virus and treated with 3% DSS. Intraperitoneal administration of rCT-NAMPT conjugated with Cy5.5 for 1 h and colon harvest (left). Colon harvests were used for IB with αITGA6, αITGB1 or αActin and analysis of the number of ITGA6+ or ITGB1+ cells was done by FACS. The data come from three independent experiments that yielded comparable results (**A**–**C**).

**Figure 6 antioxidants-11-02376-f006:**
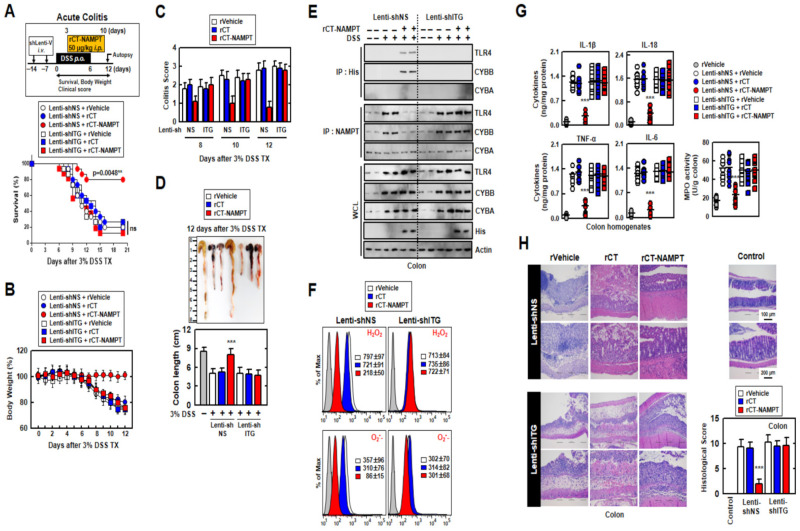
rCT-NAMPT has a medicinal effect against acute DSS-induced colitis in mice. (**A**) Scheme of the acute model of colitis transduced with Lenti-shNS or Lenti-shITG virus and subjected to 3% DSS with rCT-NAMPT (50 μg/kg) (upper). The survival of mice was monitored for 12 days; mortality was measured for *n* = 15 mice per group (lower). The statistical differences between the rVehicle-treated animals are noted (log-rank test). The data come from two different experiments that yielded comparable results. (**B**) Weight loss (*n* = 8). (**C**) Colitis scores were obtained from clinical parameters (weight loss, stool consistency, bleeding) (*n* = 8). (**D**) Image (upper) and length (lower) of colon in 3% DSS-treated mice with rVehicle, rCT or rCT-NAMPT (*n* = 8). (**E**) Colon was used for IP with αHis or αNAMPT, followed by IB with αTLR4, αCYBB or αCYBA. WCLs were used for IB with αTLR4, αCYBB, αCYBA, αHis or αActin. (**F**) FACS analysis for cellular ROS from colon (*n* = 8). (**G**) Levels of cytokines and MPO activity in colon homogenates (*n* = 10). (**H**) Hematoxylin and eosin (H&E) staining of the colon (left) (*n* = 10): representative imaging Histopathology scores were assessed in 3% DSS-treated mice with rVehicle, rCT, or rCT-NAMPT using H&E staining, as indicated in techniques (Materials and Methods). Statistical significance was assessed using the Student’s *t*-test with the Bonferroni correction (*** *p* < 0.001) in comparison to rVehicle.

**Figure 7 antioxidants-11-02376-f007:**
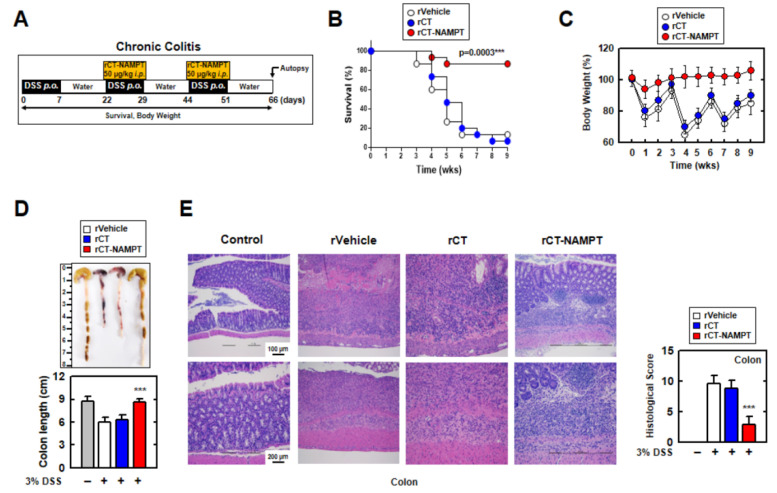
rCT-NAMPT relieves chronic DSS-induced colitis in a mouse model. (**A**) Scheme of the chronic colitis model subjected to 3% DSS with rCT-NAMPT (50 μg/kg). (**B**) Mice survival was tracked for 9 weeks, and mortality was calculated for *n* = 15 mice per group. The statistical differences between the rVehicle-treated animals are noted (log-rank test). The data come from two different experiments that yielded comparable results. (**C**) Weight loss of rVehicle, rCT or rCT-NAMPT in mice (*n* = 15). (**D**) Image (upper) and length (lower) of colon in 3% DSS-induced chronic colitis mice with rVehicle, rCT or rCT-NAMPT. (**E**) H&E staining of the colon (left) (*n* = 8): a representative image. H&E staining was used to evaluate histopathology scores in 3% DSS-induced chronic colitis mice with rVehicle, rCT, or rCT-NAMPT. The Student’s *t*-test with the Bonferroni adjustment was used to establish statistical significance when compared to rVehicle (*** *p* < 0.001).

**Table 1 antioxidants-11-02376-t001:** Datasets used for the study.

Dataset	Tissue	Condition and Number of Samples			Total Number of Cells	Cell Type Annotation
		Healthy	Inflamed	Non-Inflamed		
SCP259	Colon (LP and Epi)	12	18	18	365 K	Available
GSE182270	Colon (LP)	4	5	-	32 K	Not available

LP: lamina propria, Epi: Epithelium.

**Table 2 antioxidants-11-02376-t002:** Macrophage DEG analysis result summary for the genes involved in the NLRP3 inflammasome activation via the redox signaling pathway.

Genes	SCP259 (# Macrophages: 13888)					GSE182270 (# Macrophages: 555)				
	p.val	p.val.adj	logFC	pct.test	pct.ctrl	p.val	p.val.adj	logFC	pct.test	pct.ctrl
**NAMPT**	2.64 × 10^−13^	4.80 × 10^−09^	0.72	0.36	0.34	0.0011	1	0.32	0.88	0.82
**CYBB**	7.57 × 10^−21^	1.37 × 10^−16^	0.71	0.39	0.34	4.19 × 10^−06^	0.141	0.62	0.66	0.48
**NCF1**	3.79 × 10^−65^	6.87 × 10^−61^	−0.43 *	0.29	0.53	0.049602	1	−0.27 *	0.39	0.51
**NCF2**	7.77 × 10^−11^	1.41 × 10^−06^	0.53	0.24	0.20	0.001618	1	0.54	0.51	0.37
**NCF4**	1.7 × 10^−132^	3.1 × 10^−128^	−0.86 *	0.21	0.49	0.001578	1	−0.53 *	0.33	0.47
**P2RY6**	6.00 × 10^−87^	1.1 × 10^−82^	−0.67	0.09	0.26	2.23 × 10^−05^	0.752	−0.59	0.21	0.41
**SOD2**	6.58 × 10^−12^	1.2 × 10^−07^	0.489	0.51	0.522	5.89 × 10^−19^	1.9 × 10^−14^	1.074	0.901	0.653
**N × AT1**	0	0	1.171	0.798	0.585	3.21 × 10^−12^	1.1 × 10^−07^	0.467	0.971	0.874
**HIF1A**	1.63 × 10^−16^	2.9 × 10^−12^	0.790	0.196	0.135	0.002	1	0.624	0.48	0.358
**TXNIP**	7.71 × 10^−70^	1.4 × 10^−65^	0.985	0.455	0.312	0.025	1	−0.323 *	0.417	0.505
**TXN**	Not available					3.74 × 10^−05^	1	0.569	0.764	0.695
**NLRP3**	1.54 × 10^−05^	0.279	0.420	0.067	0.044	Not available				
**CASP1**	0.026	1	0.298	0.329	0.346	Not available				
**IL1B**	Not available					0.008	1	0.402	0.728	0.705
**NFKB1**	0.038	1	0.504	0.159	0.153	0.013	1	0.543	0.422	0.316
**TNFAIP8**	0.024	1	0.377	0.156	0.146	0.024	1	0.473	0.453	0.368
**TNFAIP8L2**	6.22 × 10^−18^	1.13 × 10^−13^	−0.306	0.131	0.212	0.001	1	−0.599	0.253	0.421
**TNFRSF1B**	3.36 × 10^−17^	6.11 × 10^−13^	0.714	0.277	0.218	0.011	1	0.611	0.436	0.368

P.val: *p*-value, P.val.adj: adjusted *p*-value, logFC: log fold change, pct.test/pct.control: ratio of the cells expressing the gene in the test and control group. * Not agree with the result from the other dataset, SCP259.

## Data Availability

The data presented in the study are available in the article.

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
