# Peer review of "Colon-Targeted eNAMPT-Specific Peptide Systems for Treatment of DSS-Induced Acute and Chronic Colitis in Mouse"

_antioxidants, 2022, doi:10.3390/antiox11122376_

Round 1
Reviewer 1 Report
The manuscript is based on the role of colon-targeted NAMPT -specific peptide in inflammatory bowel disorders.
I believe that the Authors have done a lot of experiments clearly demonstrating a real impact of NAMPT in acute and chronic colitis with a focus on macrophages.
The novelties of their work are: (i) the discovery for the first time a direct interaction of eNAMPT with NOX2 and (ii) the development of novel pharmacological tool (CT-NAMPT peptide) to be use not only in IBD models but also in other pathologies where eNAMPT plays a crucial role.
However, I believe that despite the number of in vitro, in vivo and database analysis, the article will be suitable for publication on Antioxidants only a after the following suggestions:
Major comments:
1-The authors described for the first an important involvement of NOX2(CYBB) in the mechanism of action of eNAMPT. The Authors clearly demonstrated a direct interaction with eNAMPT and CYBB by a lot of IP analysis. I believe that a such important think will be further investigated with biochemistry analysis. For example a direct biding of rNAMPT, with rTLR4 and rCYBB using SPR analysis, gel filtration complex etc. could be essential to bona fine demonstrate a direct complex
2- In cell Culture studies (eg. Fig 3) the authors used several peptides to clock NAMPT interaction with TLR4 and CYBB, I believe that a CTRL peptide will be used as in internal control.
3- About the mechanism of action in vivo, It could be useful to validate acute colitis in TLR4KO mice or CYBBKO mice to clearly validate the mechanism.
Minor comments:
- The figures are not in the correct order. Please check the order.
- Lane 35. Is better to say (e.g. anti-TNF) is not the only biologic in the treatment of IBD.
- Lane 54. Spelling for NF-KB
- Lane 319. In the text there is written Visfatin, but is not important in this part.
- Lane 388 change NAMPT expression with eNAMPT secretion or release.
- In general check the text in the manuscript to be well written and clearly.
Author Response
Þ Thanks for your kind and excellent comments. We submit a revised version of our manuscript and a point-by-point response to the reviewers’ comments. Detailed responses are described below
Major comments:
1-The authors described for the first an important involvement of NOX2(CYBB) in the mechanism of action of eNAMPT. The Authors clearly demonstrated a direct interaction with eNAMPT and CYBB by a lot of IP analysis. I believe that a such important think will be further investigated with biochemistry analysis. For example a direct biding of rNAMPT, with rTLR4 and rCYBB using SPR analysis, gel filtration complex etc. could be essential to bona fine demonstrate a direct complex.
Þ Thanks for your kind and excellent comments. We performed the Interaction Kinetic Analyses of the NAMPT with Binding Partners. The interactions of NAMPT with TLR4 or CYBB were monitored using a Fluoromax-4 spectrofluorometer (HORIBA Scientific), and were performed as previously described (Biochemistry 2005, 44, 15423–15429). Briefly, TLR4 or CYBB was labeled with BODIPY FL Iodoacetamide (ThermoFisher Scientific,Waltham, MA, USA), according to the manufacturer’s instructions. Labelled TLR4 or CYBB was excited at 350 nm, and detection was through a cutoff filter at 512 nm. Fluorescently labelled TLR4 or CYBB was titrated with unlabeled NAMPT for the kinetic analysis. The excitation and emission wavelengths used were 498 mm and 518 nm, respectively. The data obtained were fitted using the Grafit program. All fluorescence measurements were performed at 25 `C in 30 mM Tris, pH 7.4, 150 mM NaCl and 1 mM dithiothreitol.
We added the Interaction Kinetic Analyses of the NAMPT with Binding Partners in Fig. 2F. The in vitro interaction between NAMPT and TLR4 or CYBB, measured using a fluorescence binding assay with recombinant proteins and fluorescently labelled TLR4 or CYBB with NAMPT, showed an adequately high affinity (TLR4, 219 nM; CYBB, 896nM) (Fig. 2F). The NAMPT-TLR4 binding was stronger than that between NAMPT-CYBB. This is probably because the LRR domain of TLR4 that binds NAMPT is relatively larger than the extracellular domain 2 of CYBB.
2- In cell Culture studies (eg. Fig 3) the authors used several peptides to clock NAMPT interaction with TLR4 and CYBB, I believe that a CTRL peptide will be used as in internal control.
Þ We are sorry for our mistakes. We have corrected. We used the TAT peptide as a CTRL peptide and changed the Fig. 3.
3- About the mechanism of action in vivo, It could be useful to validate acute colitis in TLR4KO mice or CYBBKO mice to clearly validate the mechanism.
Þ Thanks for your kind and excellent comments. We performed whether the effect of rCT-NAMPT has a therapeutic effect against acute DSS-induced colitis in TLR4-/- and CYBB-/- mice compare with WT mice. In Supplementary Fig 4, rCT-NAMPT has a partially therapeutic effect against acute DSS-induced colitis in TLR4-/- or CYBB-/- mice compare with WT mice (Fig. S4). Therefore, recombinant multifunctional NAMPT (rCT-NAMPT) linked to the CT peptide that harbors the essential minimal residues required for binding CYBB and TLR4, may serve as a platform for developing safe and effective treatments of IBD.
Minor comments:
- The figures are not in the correct order. Please check the order.
Þ We are sorry for our mistakes. We have corrected.
- Lane 35. Is better to say (e.g. anti-TNF) is not the only biologic in the treatment of IBD.
Þ We are sorry for our mistakes. We have corrected.
- Lane 54. Spelling for NF-KB
Þ We are sorry for our mistakes. We have corrected.
- Lane 319. In the text there is written Visfatin, but is not important in this part.
Þ We are sorry for our mistakes. We have corrected.
- Lane 388 change NAMPT expression with eNAMPT secretion or release.
Þ We are sorry for our mistakes. We have corrected.
- In general check the text in the manuscript to be well written and clearly.
Þ The manuscript has been carefully reviewed by an experienced editor whose first language is English and who specializes in editing papers written by scientists whose native language is not English
Reviewer 2 Report
The manuscript of J.S. Kim et al. further investigates the role of eNAMPT in ulcerative colitis, using single cell RNA-sequencing. Furthermore, DSS colitis mouse models were used to investigate novel colon-targeted NAMPT-specific systems, such as novel CT peptides.
The authors made great efforts, however this reviewer has some remarks:
Line 2-3 Title: Should the title reference NAMPT or “eNAMPT”? Furthermore, you should include in the title the model used in the article: “…DSS colitis mouse model”.
Line 19: Does RNA-seq data indeed show levels of “eNAMPT” or is it “NAMPT”?
Line 42: Definition of NAD should be included
Line 58: It should be indicated earlier that “Visfatin” is another name of eNAMPT (as in line 44).
In relation to the method section, the authors stated that they used two datasets, SPC259 available from human cell atlas (Smilie, 2019) and GSE182270 (Uzzan et al, 2022) and, they indicate that other UC datasets (GSE162335 and GSE125527), do not contain a sufficient number of macrophage and T cells.
The first data set selected relates to the Smillie paper, demonstrating that M-like inflammatory cells and T cells expand in the mucosa of UC patients, which demonstrates a correct selection. However, in the second dataset (Uzzan paper), the mononuclear cell population is under-represented. The authors should explain the rationale choosing the second data set,
Line 250: The authors should describe in more details the set-up of the animal studies that they carried out. 2.12 describes the mouse models and induction of DSS colitis, but the authors should add the experimental intervention groups (+ controls), how long and when mice were treated, knockdown models and rationale behind chosen models, randomization, and number of mice per group. Fig. 7 describes some of it, but you should also mention in text.
In Fig 2. they should Indicate the molecular weight of detected proteins in western blots, even though they include the whole gels in the supplementary figure 4 and 5. Line 368: Specify the meaning of WCL (whole cell lysate) in the legend of figure 2
Line 314: Here it is explained the CYBB is also known as NOX2, this has to be established earlier on, in the introduction.
Also in line 315, the authors stated that “macrophage is the main source of chronic inflammation in IBD”. This a very broad statement, and needs to be better defined.
Line 374-376: The authors textually stated: “BMDMs were stimulated with LPS (100 ng/mL) for the indicated times (left) and primed with LPS (100 ng/mL) for 4 h and stimulated with ATP (1 mM) at various concentrations for 1h (right)”. The figure shows multiple timepoints, you need to clarify if you stimulated with ATP for 1 hour or indeed at multiple points as the figure implies.
Line 434-440: This paragraph can be moved upwards to line 386, because LPS and ATP were previously discussed here. This would be beneficial to the reader
Line 448: Similar to previous comment: explanation about TAT-peptide earlier in the text would be beneficial for the reader. Line 405 mentions it for the first time, however it is further explained 448.
Fig 4D and 4E: Unify the use for “LPS-primed macrophages” and “LPS-primed BMDMs”
Fig 4: The authors should clarify why a different time of LPS induction (6h) was used, compared to the previous figure 2 (4 h)?
Line 508-509: The authors should specify the peptide administration protocol, if it was firstly induced a DSS-colitis and later introduced rCT-NAMPT or if they administered simultaneously. Was rCT-NMAPT also given intraperitoneal during these experiments?
Fig 7A. Clarify the timing of chronic colitis, in the figure it is indicated 21 days and in the text 16 days
Line 558. P2RY6 is mentioned in the first paragraph, but it is not established earlier, nor highlighted in the results.
Line 617-619: the phrasing is a bit confusion. A previous study is mentioned without reference. Also ‘the receptor’, which receptor is actually being referred to here?
Line 639-640. Limitations of animal models compared to human physiology is mentioned. Could the authors further elaborate/hypothesize about similarities between human and mice? Also, can the collected results of colon-targeted NAMPT-specific peptide systems be translated to human applications?
Additional remarks
- The version of the manuscript presented has some issues with the figures, which do not appear chronologically and are throughout the manuscript. Please adjust accordingly.
- The way NAMPT is described throughout the article is not entirely consistent, for example (e)NAMPT, eNAMPT, Visfatin. This can cause some confusion, for instance, fig1a mentions iNAMPT and eNAMPT, but then figure 1b just mentions NAMPT.
- Please check the text for grammar, some minor typo’s for instance, Line 539: “ by we”, Line 565: “in vivo” 2 times, Line 616: ‘theTLR4-independent’
Author Response
Þ Thanks for your kind and excellent comments. We submit a revised version of our manuscript and a point-by-point response to the reviewers’ comments. Detailed responses are described below
The authors made great efforts, however this reviewer has some remarks:
Line 2-3 Title: Should the title reference NAMPT or “eNAMPT”? Furthermore, you should include in the title the model used in the article: “…DSS colitis mouse model”.
Þ We are sorry for our mistakes. We have corrected the title.
Line 19: Does RNA-seq data indeed show levels of “eNAMPT” or is it “NAMPT”?
Þ We are sorry for our mistakes. The RNA-seq data show levels of NAMPT. Levels of NAMPT were observed in assays in cells, mouse colonic tissue, and patient samples. However, through in vitro macrophage experiments (Fig. 2), it was found that the increase in iNAMPT induces an increase in eNAMPT and is correlated. Therefore, it can be considered that the increase in iNAMPT observed in the RNA-seq data would also induce the increase in eNAMPT.
Line 42: Definition of NAD should be included.
Þ We are sorry for our mistakes. We have corrected.
Line 58: It should be indicated earlier that “Visfatin” is another name of eNAMPT (as in line 44).
Þ We are sorry for our mistakes. We have corrected.
In relation to the method section, the authors stated that they used two datasets, SPC259 available from human cell atlas (Smilie, 2019) and GSE182270 (Uzzan et al, 2022) and, they indicate that other UC datasets (GSE162335 and GSE125527), do not contain a sufficient number of macrophage and T cells.
The first data set selected relates to the Smillie paper, demonstrating that M-like inflammatory cells and T cells expand in the mucosa of UC patients, which demonstrates a correct selection. However, in the second dataset (Uzzan paper), the mononuclear cell population is under-represented. The authors should explain the rationale choosing the second data set,
Þ Thanks for your kind and excellent comments. As the reviewer pointed, the paper by Uzzan et al, 2022 was mostly on B cells dysregulation in inflamed mucosa. Although the majority in GSE182270 was B cells, it also contains T cells and myeloid cell populations as well (Based on our cell type annotation, we identified 1.1K of macrophages, 6.9K of T cells in the datasets and the number of macrophages in other datasets we did not use was even less than those in GSE182270). Anyhow, since the number of macrophages was limited in GSE182270, we used it only supplementary for the analysis, i.e., as we mentioned in “Cell-cell interaction analysis using CellPhoneDB” subsection, GSE182270 was used only for DEG and GSEA to see if the results from SCP259 can be verified/reproduced in other datasets too. As shown in Table 1, the DEG results from SCP259 were statistically significant, while those from GSE182270 were not due to the limited number of cells. However, the tendencies in fold changes were mostly like each other, even though they were not completely matched. To emphasize the supplementary usage of the second dataset, we added a short paragraph at the end of the first paragraph in “Analysis of public single-cell RNA-seq data for ulcerative colitis” subsection.
“Since the number of macrophages is limited in GSE182270, we used it only supplementary for DEG and GSE analysis to verify the main results from SCP259.”
Line 250: The authors should describe in more details the set-up of the animal studies that they carried out. 2.12 describes the mouse models and induction of DSS colitis, but the authors should add the experimental intervention groups (+ controls), how long and when mice were treated, knockdown models and rationale behind chosen models, randomization, and number of mice per group. Fig. 7 describes some of it, but you should also mention in text.
Þ We are sorry for our mistakes. We have corrected as below,
Acute colitis model transduced with Lenti-shNS or Lenti-shITG virus (1x1011 pfu/kg) on days -7 and -14 via i.v. before DSS treatment. While the mice were treated with 3% DSS for 6 days, rVehicle, rCT or rCT-NAMPT (50 μg/kg) was i.p. injected 8 times. The survival of mice was monitored for 12 days; mortality was measured for n = 15 mice per group. The survival of mice model transduced with Lenti-shNS has similar effects to WT control. Chronic colitis model treated with 3% DSS for 7 days at 3 cycles and water for 14 days in the interval of the cycle. The rVehicle, rCT or rCT-NAMPT (50 μg/kg) was i.p. injected at 2 cycles together with DSS treatment. The survival of mice was monitored for 9 weeks; mortality was measured for n = 15 mice per group. To cal-culate causality, account for perturbations, and reduce bias, a randomization method is used to randomly assign mice to either a treatment group or a control group (or multi-ple intervention groups).
In Fig 2. they should Indicate the molecular weight of detected proteins in western blots, even though they include the whole gels in the supplementary figure 4 and 5. Line 368: Specify the meaning of WCL (whole cell lysate) in the legend of figure 2.
Þ We are sorry for our mistakes. We have corrected.
Line 314: Here it is explained the CYBB is also known as NOX2, this has to be established earlier on, in the introduction.
Þ We are sorry for our mistakes. We have corrected in Abstract and Introduction.
Also in line 315, the authors stated that “macrophage is the main source of chronic inflammation in IBD”. This a very broad statement, and needs to be better defined.
Þ We are sorry for our mistakes. We have corrected.
Line 374-376: The authors textually stated: “BMDMs were stimulated with LPS (100 ng/mL) for the indicated times (left) and primed with LPS (100 ng/mL) for 4 h and stimulated with ATP (1 mM) at various concentrations for 1h (right)”. The figure shows multiple timepoints, you need to clarify if you stimulated with ATP for 1 hour or indeed at multiple points as the figure implies.
Þ We are sorry for our mistakes. We have corrected the figure legends.
Line 434-440: This paragraph can be moved upwards to line 386, because LPS and ATP were previously discussed here. This would be beneficial to the reader.
Þ Thanks for your kind and excellent comments. We rearranged the text as reviewer 2 comments.
Line 448: Similar to previous comment: explanation about TAT-peptide earlier in the text would be beneficial for the reader. Line 405 mentions it for the first time, however it is further explained 448.
Þ Thanks for your kind and excellent comments. We rearranged the text as reviewer 2 comments.
Fig 4D and 4E: Unify the use for “LPS-primed macrophages” and “LPS-primed BMDMs”
Þ We are sorry for our mistakes. We have corrected.
Fig 4: The authors should clarify why a different time of LPS induction (6h) was used, compared to the previous figure 2 (4 h)?
Þ We are sorry for our mistakes. We have corrected the time.
Line 508-509: The authors should specify the peptide administration protocol, if it was firstly induced a DSS-colitis and later introduced rCT-NAMPT or if they administered simultaneously. Was rCT-NMAPT also given intraperitoneal during these experiments?
Þ We are sorry for our mistakes. We have corrected the protocols.
Fig 7A. Clarify the timing of chronic colitis, in the figure it is indicated 21 days and in the text 16 days
Þ We are sorry for our mistakes. We have corrected.
Line 558. P2RY6 is mentioned in the first paragraph, but it is not established earlier, nor highlighted in the results.
Þ Thanks for your kind and excellent comments. We have corrected.
Line 617-619: the phrasing is a bit confusion. A previous study is mentioned without reference. Also ‘the receptor’, which receptor is actually being referred to here?
Þ Thanks for your kind and excellent comments. We have added the References and corrected.
Line 639-640. Limitations of animal models compared to human physiology is mentioned. Could the authors further elaborate/hypothesize about similarities between human and mice? Also, can the collected results of colon-targeted NAMPT-specific peptide systems be translated to human applications?
Þ Thanks for your kind and excellent comments. We have added in Discussion as below,
Our results show that the NAMPT, TLR4, and CYBB genes associated with mouse colitis share functions with those of humans. It is believed that molecular biology and immunological results from mouse model systems can be translated into insights into human biology and health through interpretation based on mouse experimental data.
Additional remarks
- The version of the manuscript presented has some issues with the figures, which do not appear chronologically and are throughout the manuscript. Please adjust accordingly.
Þ We are sorry for our mistakes. We have corrected.
- The way NAMPT is described throughout the article is not entirely consistent, for example (e)NAMPT, eNAMPT, Visfatin. This can cause some confusion, for instance, fig1a mentions iNAMPT and eNAMPT, but then figure 1b just mentions NAMPT.
Þ We are sorry for our mistakes. We have corrected.
- Please check the text for grammar, some minor typo’s for instance, Line 539: “ by we”, Line 565: “in vivo” 2 times, Line 616: ‘theTLR4-independent’
Þ We are sorry for our mistakes. We have corrected.
